# PD-Diag-Net: Clinical-Priors guided Network on Brain MRI for Auxiliary Diagnosis of Parkinson's Disease

## Abstract

Parkinson's disease (PD) is a common neurodegenerative disorder that severely diminishes patients' quality of life. Its global prevalence has increased markedly in recent decades. Current diagnostic workflows are complex and heavily reliant on neurologists' expertise, often resulting in delays in early detection and missed opportunities for timely intervention. To address these issues, we propose an end-to-end automated diagnostic method for PD, termed **PD-Diag-Net**, which performs risk assessment and auxiliary diagnosis directly from raw MRI scans. This framework first introduces an MRI Pre-processing Module (**MRI-Processor**) to mitigate inter-subject and inter-scanner variability by flexibly integrating established medical imaging preprocessing tools. It then incorporates two forms of clinical prior knowledge: (1) Brain-Region-Relevance-Prior (**Relevance-Prior**), which specifies brain regions strongly associated with PD; and (2) Brain-Region-Aging-Prior (**Aging-Prior**), which reflects the accelerated aging typically observed in PD-associated regions. Building on these priors, we design two dedicated modules: the Relevance-Prior Guided Feature Aggregation Module (**Aggregator**), which guides the model to focus on PD-associated regions at the inter-subject level, and the Age-Prior Guided Diagnosis Module (**Diagnoser**), which leverages brain age gaps as auxiliary constraints at the intra-subject level to enhance diagnostic accuracy and clinical interpretability. Furthermore, we collected external test data from our collaborating hospital. Experimental results show that PD-Diag-Net achieves 86% accuracy on external tests and over 96% accuracy in early-stage diagnosis, outperforming existing advanced methods by more than 20%.

## 1 Introduction

**Background.** Parkinson's disease (PD) is a common neurodegenerative disorder, and its global prevalence has increased dramatically in recent decades, with the number of patients rising from approximately 2.5 million in 1990 to 6.3 million in 2016, and projected to surpass 12 million by 2040 (Bloem et al., 2021; Dorsey et al., 2018; Rocca, 2018). PD progresses slowly yet is highly disabling, with common symptoms including sleep disturbances, olfactory dysfunction, autonomic nervous system abnormalities, emotional and cognitive impairment, speech and swallowing difficulties, as well as chronic fatigue and pain, all of which severely affect patients' quality of life.

**Motivation.** PD currently has no cure, but early detection and timely pharmacological intervention can significantly slow disease progression and alleviate motor symptoms. However, the current diagnostic workflow poorly suited to achieve this goal, which can be briefly summarized as follows (see Appendix D for the full diagnostic flowchart): (1) Neurologists conduct a preliminary assessment based on the patient's medical history and hallmark motor symptoms (*e.g.*, bradykinesia, rigidity). (2) For suspected cases, magnetic resonance imaging (MRI) is used to exclude confounding conditions (*e.g.*, stroke, brain tumors) that may present with similar symptoms, followed by PD confirmation. This highly expertise-dependent diagnostic process has clear drawbacks: (1) For clinicians, it increases workload, lacks objective imaging biomarkers, and limits diagnostic consistency. (2) For patients, it requires them to seek care from top-tier medical centers, increasing their financial burden and potentially causing delays that result in missed opportunities for the optimal treatment window.

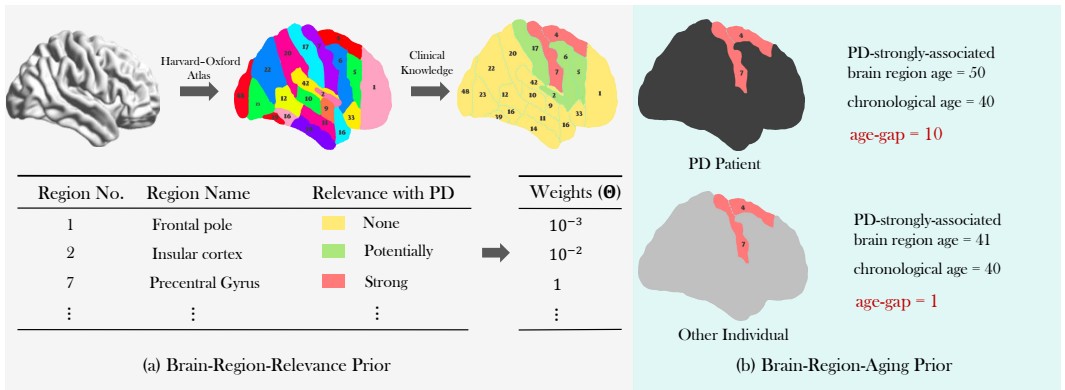

Figure 1: **(a) Brain-Region-Relevance-Prior (Relevance-Prior)**. The human brain can be functionally parcellated into distinct regions. In this study, we employ the Harvard–Oxford Atlas (Jenkinson et al., 2012) to divide the brain into 48 regions (see Appendix E). Based on clinical prior knowledge, these regions are categorized as strongly-associated (red), potentially-associated (yellow), or non-associated (green) with Parkinson's disease (PD), and are assigned weights of $1$, $10^{-2}$, and $10^{-3}$, respectively. This prior guides the model in capturing inter-subject level features. **(b) Brain-Region-Aging-Prior (Aging-Prior)**. The human brain undergoes progressive decline with aging. For PD patients, however, strongly-associated regions show accelerated aging, leading to a larger brain age compared to chronological age. In contrast, non-PD individuals (including healthy controls and other neurological disorders) show no such accelerated decline in PD-strongly-associated regions, resulting in smaller brain age gaps. This prior helps characterize intra-subject level differences.

Therefore, we are committed to developing a method for PD risk assessment directly from MRI scans. It is designed to enable individuals to leverage raw MRI from regular health checkups for self-screening, early risk detection, and timely medical consultation. At the same time, it provides clinicians with an objective and interpretable decision-support tool to reduce workload, improve consistency, and complement existing diagnostic workflows.

**Technical Challenge.** To achieve this goal, we analyzed real-world data, including public datasets and raw MRI scans we collected from clinical settings, and identified two key challenges: (Challenge-A) Substantial variability in brain morphology, size, and signal characteristics across individuals, along with differences in MRI scanners and acquisition protocols, results in highly heterogeneous data distributions that limit model generalization. (Challenge-B) The whole-brain MRI differences between PD patients and healthy individuals are not clearly distinguishable, making it challenging even for experienced neurologists to diagnose PD based solely on MRI (which explains why MRI is primarily used to rule out other conditions rather than confirm PD). In parallel, we reviewed existing approaches and found that most studies (Islam et al., 2024; Alrawis et al., 2025) are evaluated only on narrowly defined, heavily preprocessed datasets, lacking adaptability to the complexities of real-world practice and thus limiting their practical applicability. These findings underscore the urgent need for new algorithms to address these challenges.

**Solution.** In this paper, we propose an end-to-end PD Diagnosis Network (**PD-Diag-Net**, please see Fig. 2 for the detailed flowchart), which leverages mature medical imaging toolkits and incorporates clinical prior knowledge to enable automated and interpretable PD prediction.

To address Challenge-A, we design an MRI Pre-processing Module (**MRI-Processor**, see Appendix F) that employs Highly Accurate Deep Brain Extraction Tool (HD-BET) (Isensee et al., 2019) and Advanced Normalization Tools (ANTs) (Avants et al., 2009) to perform skull stripping (removing skull and non-brain tissues), bias field correction, and nonlinear registration (standardizing brain size and alignment) on raw T1-weighted MRI. These steps greatly reduce the adverse effects of data inconsistency and provide a solid foundation for robust, generalizable modeling.

In response to Challenge-B, we incorporate two forms of clinical prior knowledge, enabling the model to better capture differences between PD patients and other subjects (healthy controls and individuals with neurological disorders) from individual-level MRI data. (1) Brain-Region-Relevance-Prior (**Relevance-Prior**, see Fig. 1 left): The human brain can be functionally parcellated into multiple regions, and this prior defines each region's relevance to PD. Building on this knowledge, we propose the Relevance-Prior Guided Feature Aggregation Module (**Aggregator**, see Fig. 3 left) at

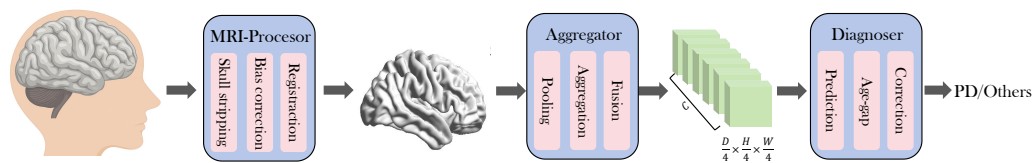

Figure 2: Flowchart of the Parkinson's Disease Diagnosis Network (**PD-Diag-Net**), consists of 3 modules: MRI Pre-processing Module (**MRI-Processor**, see Appendix F), Relevance-Prior Guided Feature Aggregation Module (**Aggregator**, see Fig. 3, left), and Age-Prior Guided Diagnosis Module (**Diagnoser**, see Fig. 3, right).

the inter-subject level (*i.e.*, across individuals), which applies brain-region-wise average pooling to focus on PD-associated regions' features, thereby improving both the model's discriminative capability and interpretability. (2) Brain-Region-Aging-Prior (**Aging-Prior**, see Fig. 1 right): PD patients typically exhibit accelerated aging in PD-associated brain regions, meaning the brain age of these regions is often considerably higher than the subject's chronological age, whereas other subjects generally show minimal or no such gap. Based on this observation, we develop the Aging-Prior Guided Parkinson's Disease Diagnosis Module (**Diagnoser**, see Fig. 3 right) at the intra-subject level (*i.e.*, within a single brain), which incorporates the brain age gap as an auxiliary signal to constrain classification logits, further enhancing diagnostic accuracy and clinical interpretability.

**Contribution.** (1) We propose the end-to-end PD-Diag-Net, consisting of three core modules (MRI-Processor, Aggregator, Diagnoser) and designed to handle raw MRI data from diverse sources and acquisition protocols, laying a solid technical foundation for early PD screening and intervention. (2) We systematically integrate two types of brain-region-related clinical prior knowledge into the model design, offering insights for automated imaging-based diagnosis of other neurological disorders. (3) To comprehensively evaluate the model's real-world performance, we trained it on publicly available PD datasets and further conducted external test using a self-collected dataset (to be publicly released). Results show that our model achieves over 86% accuracy, substantially outperforming existing methods. Moreover, in a dedicated evaluation on early-stage PD cases, the model attains 96% accuracy, highlighting its strong potential for early detection.

## 2 RELATED WORK

Currently, AI-assisted diagnosis of PD can be approached from two perspectives. On the one hand, there is the behavioral perspective based on clinical symptoms. Clinicians typically rely on observable symptoms to determine whether a subject has PD. Accordingly, most existing AI-assisted methods are designed to quantify such behaviors from the clinician's standpoint. For instance, (Talitckii et al., 2022; Aouraghe et al., 2023; Wang et al., 2024) analyze handwriting patterns, (Thies et al., 2025; Xu et al., 2025; Favaro et al., 2024) evaluate speech characteristics, and (Navita et al., 2025; Tang et al., 2024; Zhang et al., 2024) assess gait dynamics. **However, these approaches are effective only when symptoms are already evident, meaning the disease has progressed to a relatively advanced stage, which limits their utility for early detection and intervention.**

On the other hand, there is the imaging perspective. To the best of our knowledge, research on MRI-based PD diagnosis remains limited. Existing studies (Dentamaro et al., 2024; Alrawis et al., 2025; Erdaş & Sümer, 2023; Islam et al., 2024) directly apply classical computer vision models to this domain, typically focusing on performance within narrowly defined and heavily preprocessed datasets. Such models are difficult to deploy in real-world clinical settings, suffer from poor interpretability, and rarely release their code, further restricting reproducibility and clinical impact.

## 3 METHODOLOGY

### 3.1 OVERVIEW

Our proposed **Parkinson's Disease Diagnosis Network (PD-Diag-Net)** comprises three modules: MRI Pre-processing Module (**MRI-Processor**), Relevance-Prior Guided Feature Aggregation Module (**Aggregator**), and Aging-Prior Guided Parkinson's Disease Diagnosis Module (**Diagnoser**).

**MRI-Processor** (see Appendix F) integrates advanced MRI processing tools to standardize raw T1-weighted MRI scans, ensuring data quality and consistency while minimizing adverse effects caused by differences in scanner hardware and individual brain anatomy. It consists of three key steps: **(1)** Employ the HD-BET (Isensee et al., 2019) to automatically remove the skull and non-brain tissues, isolating the brain parenchyma and facilitating subsequent analysis. **(2)** Apply the Nonparametric Nonuniform intensity Normalization (N4) algorithm in ANTs (Avants et al., 2009) to correct bias fields, reducing intensity artifacts caused by magnetic field inhomogeneities during MRI acquisition and improving overall intensity uniformity. **(3)** Use the Symmetric Normalization (SyN) algorithm in ANTs to nonlinearly register individual MRI scans to the Montreal Neurological Institute (MNI) standard template, enabling consistent cross-subject comparison and group-level analysis.

**Aggregator** (see Fig. 3 left) incorporates Relevance-Prior to guide the model's focus toward brain regions strongly associated with PD, further mitigating adverse effects of irrelevant or noisy information. It consists of four main steps: **(1)** Use the 3D DenseNet (Ruiz et al., 2020) to perform early-stage dense encoding on MRI scans registered to the standard space, generating whole-brain feature representations. **(2)** Perform brain-region-wise average pooling based on the raw MRI and Harvard–Oxford Atlas (Jenkinson et al., 2012) to generate region-level features. **(3)** Incorporate clinically informed prior weights to emphasize features from brain regions highly relevant to PD, and perform a weighted aggregation of these region-level features. **(4)** Reshape the aggregated representation to match the spatial dimensions of the dense encoding, and fuse it with the dense feature.

**Diagnoser** (see Fig. 3 right) leverages the Aging-Prior knowledge to constrain diagnostic outcomes and improve prediction accuracy. This module consists of two parallel branches and three sequential steps: **(1)** Branch-1 encodes the fusion feature representation and performs classification to determine whether the subject has PD. **(2)** Branch-2 utilizes an identical network architecture (with independently trained parameters) to encode the same fusion feature representation and predict the age of PD-associated brain regions. **(3)** Compute the brain age of PD-associated regions and compare it with the chronological age; a larger age gap suggests a higher likelihood of PD, and this measure serves as an auxiliary constraint to refine diagnostic predictions.

### 3.2 Clinical Prior Knowledge about Parkinson's Disease

**Brain-Region-Relevance-Prior** (**Relevance-Prior**, see Fig. 1 left). The human brain can be divided into multiple regions based on functional characteristics, and in this study, we adopt the Harvard–Oxford Atlas (Jenkinson et al., 2012) for standardized brain parcellation. PD is a neurodegenerative disorder whose pathological changes are not uniformly distributed across the entire brain; instead, pronounced abnormalities primarily appear in specific regions responsible for motor control, executive function, and emotional regulation. Drawing on clinical expertise and previous research findings (Gao & Wu, 2016; Burciu & Vaillancourt, 2018), we categorize the association between each brain region and PD into three levels: strongly-associated, non-associated, and potentially-associated. Based on this clinically informed prior knowledge, we assign differentiated weights to each brain region and incorporate these weights into the feature modeling process to emphasize PD-relevant patterns, suppress irrelevant signals, and ultimately improve both diagnostic accuracy and model interpretability. We define the atlas as $\mathbf{M}_{\text{raw}} \in \mathbb{R}^{D \times H \times W}$, where each voxel is assigned an integer label from $0$ to $R$, corresponding to one of $R$ brain regions, *i.e.,* $\mathbf{M}_{\text{raw}}[i,j,k] \in \{0,1,2,\cdots,R\}$, $D, H, W$ represent the depth, height, and width of the atlas. The prior weights derived from clinical expertise are denoted as $\boldsymbol{\Theta} \in \mathbb{R}^{R}$, where each element $\boldsymbol{\Theta}[r]$ indicates the clinical relevance of the $r$-th brain region.

**Brain-Region-Aging-Prior** (**Aging-Prior**, see Fig. 1 right). As humans age, the entire brain undergoes a general aging process. However, clinical experience and previous studies (Sarasso et al., 2021; Liu et al., 2020) have shown that patients with PD exhibit significantly accelerated aging in certain PD-associated brain regions, meaning that the brain age of these regions is often substantially higher than the subject's chronological age. Based on this prior knowledge, we calculate the age gap between the predicted brain age of PD-associated regions and the subject's chronological age; a larger age gap indicates a higher likelihood of PD. We then incorporate this brain age gap as a diagnostic constraint to help the model more accurately distinguish PD patients from healthy controls or individuals with other neurological conditions, thereby improving diagnostic accuracy and clinical interpretability. Define the predicted brain age of PD-associated regions as $\hat{A}_{\text{pd}}$, and the

Figure 3: In **Aggregator** (left), the pre-processed MRI is first passed through a 3D DenseNet to extract dense features. The MRI and the brain region atlas are then used to perform brain-region-wise average pooling, followed by weighted aggregation to compute the mean and variance of PD-associated regions. The aggregated features are subsequently upsampled to match the dimensions of the dense features and fused together. In **Diagnoser** (right), the fused features are fed into both Branch-1 and Branch-2: Branch-1 outputs the prediction logits for PD, while Branch-2 estimates the age gap between PD-associated regions and the subject's chronological age. This age gap is then used to constrain the prediction logits, yielding the final classification.

known chronological age as $A_{\text{chrono}}$, the age gap $\Delta$ can be formulated as:

$$\Delta = \hat{A}_{\text{pd}} - A_{\text{chrono}}. \tag{1}$$

### 3.3 MRI Pre-processing Module

Due to variations in resolution, contrast, and signal-to-noise ratio across MRI scanners, as well as substantial inter-subject differences in brain size, shape, and anatomical structure, we propose MRI-Processor to standardize all raw T1-weighted MRI scans through a three-step pre-processing pipeline to enhance the robustness of downstream modeling and improve cross-subject comparability: (1) Brain extraction: Remove the skull and non-brain tissues to isolate the brain parenchyma, reducing irrelevant background noise; (2) Bias field correction: Correct intensity inhomogeneities caused by magnetic field non-uniformities, improving overall image intensity uniformity and comparability; and (3) Nonlinear registration: Align each MRI scan to the standard MNI template to reduce anatomical variability between subjects and facilitate cross-subject analysis and model generalization. This entire workflow was implemented using mature open-source toolkits. As different tools excel at specific tasks, we adopt the following combination to balance computational efficiency and output quality: HD-BET (Isensee et al., 2019) for brain extraction, the N4 algorithm in ANTs (Avants et al., 2009) for bias field correction, and the SyN algorithm in ANTs (Avants et al., 2009) for nonlinear registration. Define the raw MRI image as $\mathbf{X}_{\text{raw}} \in \mathbb{R}^{D \times H \times W}$, the workflow is:

$$\mathbf{X}_{\text{proc}} = \mathcal{F}_{\text{SyN}} \left( \mathcal{F}_{\text{N}_4} \left( \mathcal{F}_{\text{BET}} \left( \mathbf{X}_{\text{raw}} \right) \right) \right), \tag{2}$$

where $\mathbf{X}_{\text{proc}} \in \mathbb{R}^{D \times H \times W}$ denotes the processed MRI data; $\mathcal{F}_{\text{BET}}$, $\mathcal{F}_{\text{N}_4}$, and $\mathcal{F}_{\text{SyN}}$ denote the operation of brain extraction, bias field correction and nonlinear registration to the MNI space.

### 3.4 Relevance-Prior Guided Feature Aggregation Module

Aggregator incorporates Relevance-Prior knowledge by introducing a brain-region-wise average pooling operation followed by region-weighted feature aggregation, enabling the model to emphasize features in PD-associated regions while suppressing irrelevant signals from unrelated areas. This design enhances both discriminative performance and clinical interpretability by explicitly integrating prior knowledge into the feature modeling process. The pipeline includes four steps:

(1) We first employ 3D CNN to extract dense features from the pre-processed MRI images, and then resample the atlas to match the spatial resolution of these features. We formulate the step as:

$$\mathbf{X}_{\text{dense}} = \mathcal{F}_{\text{cnn-1}} \left( \mathbf{X}_{\text{proc}} \right), \quad \mathbf{M}_{\text{proc}} = \mathcal{F}_{\text{onehot}} \left( \mathbf{M}_{\text{raw}} \right), \tag{3}$$

where $\mathcal{F}_{\text{cnn-1}}$, $\mathcal{F}_{\text{onehot}}$ denote the operation of 3D convolution and one-hot encoding of atlas labels; $\mathbf{X}_{\text{dense}} \in \mathbb{R}^{C \times \frac{D}{4} \times \frac{H}{4} \times \frac{W}{4}}$, $\mathbf{M}_{\text{proc}} \in \mathbb{R}^{R \times D \times H \times W}$ denote the dense feature and one-hot encoded atlas.

(2) Next, we perform brain-region-wise average pooling to aggregate processed MRI into region-level representations, which can be formulated as:

$$\mathbf{X}_{\mathrm{pool}}[r] = \frac{\sum_{(d,h,w)\in\Omega_r} \mathbf{X}_{\mathrm{proc}}[d,h,w]}{|\Omega_r|}, \tag{4}$$

where $\mathbf{X}_{\mathrm{pool}} \in \mathbb{R}^{R\times1\times1\times1}$ denotes the pooling feature; $\Omega_r$ is the voxel set belonging to region $r$.

(3) Then, we introduce the clinical relevance weights $\boldsymbol{\Theta}$ to compute the weighted mean ($\mathbf{X}_{\mathrm{mean}} \in \mathbb{R}^{1\times1\times1}$) and standard deviation ($\mathbf{X}_{\mathrm{std}} \in \mathbb{R}^{1\times1\times1}$) of the PD-associated brain regions, which are then concatenated to form the aggregated feature ($\mathbf{X}_{\mathrm{agg}} \in \mathbb{R}^{2\times1\times1}$). We formulate the step as:

$$\mathbf{X}_{\mathrm{agg}} = \big[\mathbf{X}_{\mathrm{mean}},\ \mathbf{X}_{\mathrm{std}}\big] = \Big[\frac{\sum_{r=1}^{R} \boldsymbol{\Theta}[r] \cdot \mathbf{X}_{\mathrm{pool}}[r]}{\sum_{r=1}^{R} \boldsymbol{\Theta}[r]},\ \sqrt{\frac{\sum_{r=1}^{R} \boldsymbol{\Theta}[r] \cdot \big(\mathbf{X}_{\mathrm{pool}}[r] - \mathbf{X}_{\mathrm{mean}}\big)^2}{\sum_{r=1}^{R} \boldsymbol{\Theta}[r]}}\Big]. \tag{5}$$

(4) Subsequently, we upsample the aggragated feature to match the spatial dimensions of dense feature and fuse them. The process can be formulated as:

$$\mathbf{X}_{\mathrm{up}} = \mathcal{F}_{\mathrm{upsample}}\left(\mathbf{X}_{\mathrm{agg}}\right),\ \ \mathbf{X}_{\mathrm{fuse}} = \mathbf{X}_{\mathrm{up}} + \mathbf{X}_{\mathrm{dense}}, \tag{6}$$

where $\mathbf{X}_{\mathrm{up}} \in \mathbb{R}^{C\times\frac{D}{4}\times\frac{H}{4}\times\frac{W}{4}}$, $\mathbf{X}_{\mathrm{fuse}} \in \mathbb{R}^{C\times\frac{D}{4}\times\frac{H}{4}\times\frac{W}{4}}$ denotes the upsampled and fusion feature; $\mathcal{F}_{\mathrm{upsample}}$ denotes the operation of upsampling.

## 3.5 Aging-Prior Guided Parkinson's Disease Diagnosis Module

Diagnoser integrates Aging-Prior knowledge with a two-branch design. Branch-1 performs two-way classification (PD / Others) based on the fusion features produced by Aggregator. Branch-2 predicts the brain age of PD-associated regions and computes the age gap with the subject's chronological age. This age-gap-signal is then additively fused into the classification logits to impose an explicit prior-guided constraint, thereby improving discriminative performance and clinical interpretability. The procedure comprises three steps:

(1) Branch-1 first uses the 3D CNN to encode the fusion feature, followed by a classification head to perform prediction, which can be formulated as:

$$\mathbf{z} = (z_{\mathrm{pd}}, z_{\mathrm{ot}}) = \mathcal{F}_{\mathrm{cls}}\left(\mathcal{F}_{\mathrm{cnn\text{-}2}}\left(\mathbf{X}_{\mathrm{fuse}}\right)\right), \tag{7}$$

where $\mathcal{F}_{\mathrm{cnn\text{-}2}}$, $\mathcal{F}_{\mathrm{cls}}$ denote the operation of 3D convolution, classification; $\mathbf{z}$ denots the logits.

(2) Branch-2 uses the same architecture as Branch-1 but with independently trained parameters to encode the fusion feature, followed by a regression head to predict the brain age of PD-associated regions, which can be formulated as:

$$\hat{A}_{\mathrm{pd}} = \mathcal{F}_{\mathrm{reg}}\left(\mathcal{F}_{\mathrm{cnn\text{-}3}}\left(\mathbf{X}_{\mathrm{fuse}}\right)\right), \tag{8}$$

where $\mathcal{F}_{\mathrm{cnn\text{-}3}}$ and $\mathcal{F}_{\mathrm{reg}}$ denote the operation of 3D convolution and regression.

(3) Since the ground-truth brain age of PD-associated regions for PD patients is unavailable, we cannot directly compute a regression loss for this branch. To address this issue, we design an auxiliary loss that leverages prior constraints on the age gap to indirectly optimize Branch-2 and guide its parameter updates. We formulate this process as:

$$\mathcal{L}_{\mathrm{age}} = \mathbb{1}_{(y=\mathrm{pd})}\max(0, \zeta - \Delta) + \mathbb{1}_{(y\neq\mathrm{pd})}\max(0, \Delta - \tau), \tag{9}$$

where $y$ denotes the ground-truth class label; $\mathbb{1}_{(\mathrm{condition})}$ is the indicator function, equal to 1 if the condition is true and 0 otherwise; $\zeta$ and $\tau$ are hyperparameters representing the minimum acceptable age gap for PD samples and the maximum acceptable age gap for non-PD samples, respectively. **In simple understanding, the loss encourages the model to increase the age gap of PD samples to at least $\zeta$ while keeping the age gap of non-PD samples below $\tau$.** The term $[\mathbb{1}_{(y=\mathrm{pd})}\max(0, \zeta - \Delta)]$ enforces that PD samples should have an age gap no smaller than $\zeta$, if $\Delta \geq \zeta$, this penalty is 0, otherwise, the loss increases proportionally to $(\zeta - \Delta)$. Similarly, $[\mathbb{1}_{(y\neq\mathrm{pd})}\max(0, \Delta - \tau)]$ enforces that non-PD samples should have an age gap no greater than $\tau$, if $\Delta \leq \tau$, this term is 0, otherwise, the loss increases proportionally to $(\Delta - \tau)$.

Subsequently, we constrain the classification logits ($\tilde{\mathbf{z}} = (\tilde{z}_{\mathrm{pd}}, \tilde{z}_{\mathrm{ot}})$) as follows:

$$\phi(\Delta) = \mathrm{softplus}(\Delta - \tau) - \mathrm{softplus}(\tau - \Delta), \;\; \tilde{z}_{\mathrm{pd}} = z_{\mathrm{pd}} + \alpha \cdot \phi(\Delta), \;\; \tilde{z}_{\mathrm{ot}} = z_{\mathrm{ot}} - \alpha \cdot \phi(\Delta), \quad (10)$$

where $\mathrm{softplus}(x) = ln(1 + e^x)$ is a smooth variant of the ReLU function, which maps inputs to positive values while maintaining numerical stability; $\alpha$ is non-negative hyperparameter. **The intuition behind this formulation is that a larger age gap increases the PD logit while decreasing the other condition logits, vice versa.** We then compute the corrected classification loss as:

$$\mathcal{L}_{\mathrm{cls}} = \mathrm{CrossEntropy}(\tilde{\mathbf{z}}, y), \quad (11)$$

where $\mathcal{L}_{\mathrm{cls}}$ represents the corrected classification loss. The final loss function is:

$$\mathcal{L} = \mathcal{L}_{\mathrm{age}} + \mathcal{L}_{\mathrm{cls}} \quad (12)$$

## 4 EXPERIMENTS

**Data.** To better approximate real-world clinical applications, the data was divided into three parts: *Training and Normal Internal Test (Normal In. Test).* We collected 489 raw T1-weighted MRI scans from PPMI (Marek et al., 2011) and (Hirao et al., 2024; Redondo-Armenteros et al., 2025; Mortazavi et al., 2025), comprising 232 PD cases and 257 cases from healthy controls and patients with other neurological disorders. Five-fold cross-validation was performed. *Normal External Test (Normal Ex. Test).* We collected 188 raw T1-weighted MRI scans of PD patients from a collaborating hospital, supplemented with 200 cases of other neurological conditions from (Mueller et al., 2005), (Rieck et al., 2024; Horta et al., 2023). *Prodromal External Test (Prodromal Ex. Test).* To specifically evaluate sensitivity to early-stage PD, we extracted 33 prodromal PD cases from the 188 external PD scans, yielding an early-stage test set.

**Evaluation Metric.** To comprehensively evaluate the performance of our model, we adopt three key metrics (Appendix G.1 shows more details): *Accuracy* (*ACC*) measures the overall proportion of correct predictions and reflects general model performance; *True Positive Rate* (*TPR*, also known as Sensitivity or Recall) quantifies the proportion of true patients correctly identified and thus reflects the model's ability to capture PD cases with fewer missed diagnoses; *False Positive Rate* (*FPR*) indicates the proportion of others incorrectly classified as patients, reflecting the risk of misdiagnosis.

**Comparison Methods.** We compared our approach against several existing methods specifically developed for PD diagnosis, including XAI (Dentamaro et al., 2024), FCN-PD (Alrawis et al., 2025), FAA (Erdaş & Sümer, 2023), and SMOTE (Islam et al., 2024). In addition, we adapted a number of representative models originally designed for general brain image analysis to the PD diagnostic task, including M3T (Jang & Hwang, 2022), Swin UNETR (Tang et al., 2022), S3D (Wald et al., 2025), 3DMAE (Chen et al., 2023), and AE-FLOW (Zhao et al., 2023).

**Implementation.** Our model was trained using the AdamW optimizer, configured with an initial learning rate of $1 \times 10^{-3}$ and a weight decay of $1 \times 10^{-3}$. The learning rate schedule followed a cosine annealing strategy. Several key hyperparameters were involved in training, including $\alpha = 1$, $\zeta = 9.5$, and $\tau = 4.5$. All experiments were conducted on a high-performance computing system equipped with an NVIDIA RTX A100 GPU (48 GB memory). The training batch size was set to 4 to balance computational efficiency with optimization stability. The codes will be released publicly.

**Comparison Results and Analysis.** The comparison results with other advanced methods are summarized in Tab. 1, we observe that: (1) Most methods, even simple ones, achieve good results on internal data, indicating that when training and test distributions match, the task is relatively less challenging. Therefore, internal test performance alone cannot fully reflect clinical applicability. (2) In contrast, competing methods drop sharply in performance on external data, while only our approach maintains accuracy above 86.1%, demonstrating strong robustness and generalizability under distribution shifts across imaging centers, which is critical for real-world clinical deployment. (3) On the specially curated prodromal PD test set, our method achieves 96.7% accuracy, exceeding other approaches by more than 40%. This underscores the model's ability to capture early PD-specific imaging biomarkers and its strong potential for early screening applications. In addition, for the prodromal external test, only the TPR is reported, as the outcomes for the "Others" class, corresponding to the FPR, have already been presented in the normal external test.

| Method | Param | Normal In. Test | | | Normal Ex. Test | | | Prodromal Ex. Test |
|---|---|---|---|---|---|---|---|---|
| | | ACC (↑) | TPR (↑) | FPR (↓) | ACC (↑) | TPR (↑) | FPR (↓) | TPR (↑) |
| XAI (Dentamaro et al., 2024) | 69.8M | 94.2 | 93.0 | 5.7 | 55.3 | 62.7 | 51.7 | 53.7 |
| FCN-PD (Alrawis et al., 2025) | 10.9M | 91.7 | 92.5 | 8.4 | 56.0 | 53.2 | 41.4 | 48.5 |
| FAA (Erdaş & Sümer, 2023) | 7.3M | 90.6 | 94.2 | 9.8 | 58.5 | 66.2 | 48.7 | 47.9 |
| SMOTE (Islam et al., 2024) | 12.5M | 91.6 | 93.7 | 8.6 | 54.4 | 61.0 | 51.8 | 52.1 |
| M3T (Jang & Hwang, 2022) | 29.1M | 95.5 | 99.0 | 4.9 | 61.7 | 59.5 | 36.2 | 56.4 |
| Swin UNETR (Tang et al., 2022) | 27.1M | 96.6 | 97.5 | 3.5 | 58.5 | 63.4 | 46.1 | 53.5 |
| S3D (Wald et al., 2025) | 31.2M | 95.5 | 96.6 | 4.6 | 59.9 | 61.2 | 41.3 | 52.1 |
| 3DMAE (Chen et al., 2023) | 92.6M | 96.2 | 98.5 | 4.0 | 64.2 | 66.0 | 37.5 | 50.0 |
| AE-FLOW (Zhao et al., 2023) | 97.6M | 97.1 | 97.5 | 2.9 | 66.1 | 74.0 | 41.3 | 52.4 |
| **PD-Diag-Net (Ours)** | 89.5M | **98.5** (+1.4) | **100** (+1.0) | **1.7** (+1.2) | **86.1** (+20.7) | **93.1** (+19.1) | **20.5** (+15.7) | **96.7** (+30.3) |

Table 1: Comparison results (%). *In.* and *Ex.* denote *Internal* and *External*, respectively. ↓ indicates that lower values are better, whereas ↑ indicates that higher values are better. ROC curves are shown in Appendix G.2.

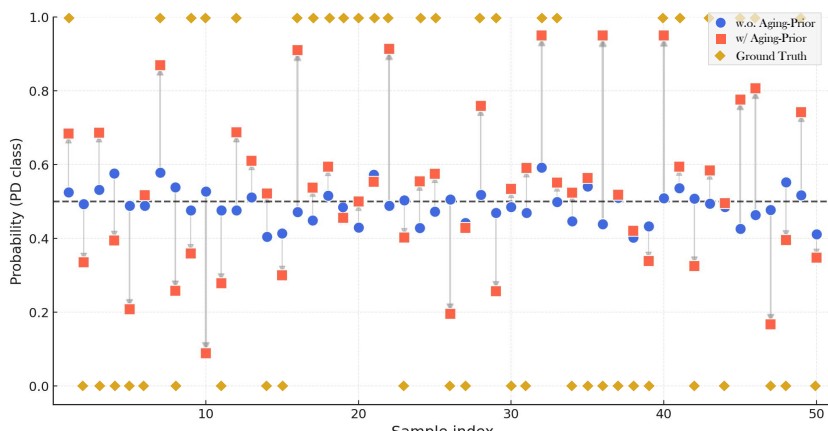

Figure 4: Efficiency of Aging-Prior on the normal external test data.

**Ablation Study.** In this study, we designed three key modules and conducted ablation studies to evaluate their individual contributions. The results are summarized in Tab. 2: (1) Row 1 represents training directly on raw MRI data. Compared with Row 5, the results clearly show that a unified pre-processing pipeline is essential, leading to an improvement of approximately 14% in accuracy. (2) Row 2 removes the Relevance-Prior and directly trains on whole-brain features. Compared with Row 5, the accuracy drops by about 25%, clearly demonstrating the necessity and importance of incorporating the Relevance-Prior into the model. (3) Row 3 uses only PD-associated brain regions while discarding all others. Compared with Row 5, the accuracy decreases slightly by about 5%, indicating that non-PD-associated regions also contain valuable auxiliary information, and completely discarding them results in a loss of discriminative power.

| Processor | Aggregator | | Diagnoser | | ACC |
|---|---|---|---|---|---|
| | $X_{up}$ | $X_{dense}$ | Bra-1 | Bra-2 | |
| | ✓ | ✓ | ✓ | ✓ | 72.4 |
| ✓ | | ✓ | ✓ | ✓ | 70.8 |
| ✓ | ✓ | | ✓ | ✓ | 81.5 |
| ✓ | ✓ | ✓ | ✓ | | 77.2 |
| ✓ | ✓ | ✓ | ✓ | ✓ | **86.1** |

Table 2: Ablation study (%) of different modules on normal external test set. *Bra* is short for Branch.

(4) Row 4 removes the Aging-Prior correction and instead applies a simple classification head. Compared with Row 5, the accuracy drops substantially by about 9%, further highlighting the critical role of the Aging-Prior in improving the model's ability to identify PD. To further validate the Aging-Prior, we examined 50 normal external test samples with predicted probabilities near 0.5 when the prior was not applied. Incorporating the Aging-Prior significantly increased the decision margin and substantially improved classification accuracy. The results are shown in Fig. 4.

**Failure Case and Analysis.** As shown in Fig. 5, we present the confusion matrix on the normal external test set and illustrate three-view (axial, coronal, sagittal) brain maps corresponding to the

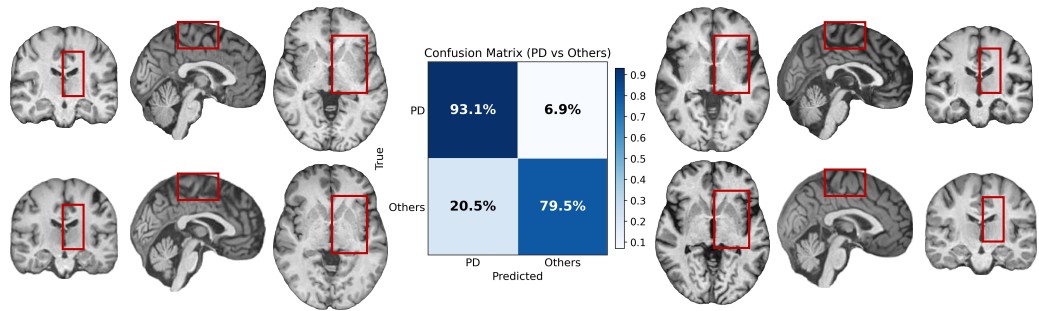

Figure 5: Confusion matrix on the normal external test set with corresponding three-view brain maps. The model achieves high accuracy in identifying PD, with most errors arising from other cases misclassified as PD.

four outcomes. The results indicate that the model performs well in identifying PD cases, with most errors arising from misclassifying non-PD cases as PD, reflecting a relatively high false positive rate. On the one hand, we acknowledge this limitation as an important direction for future work. On the other hand, in clinical practice, false positives are less critical than false negatives, since our method serves as an initial screening tool and subsequent physician evaluation can correct such errors, mitigating their impact.

To further investigate this phenomenon, we extracted and visualized PD-associated brain-region features ($\mathbf{X}_{up}$) of both PD patients and other cases from the normal external test set (Fig. 6). The results reveal that PD patients exhibit highly consistent feature distributions in these regions, *i.e.*, once PD is present, the corresponding brain-region features tend to converge,

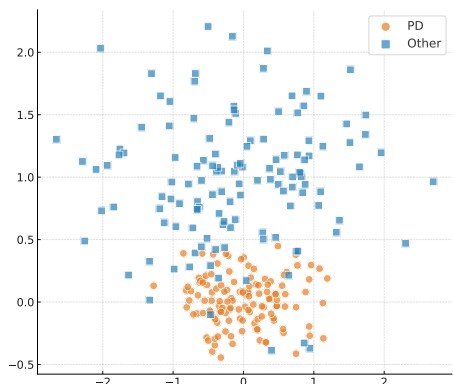

Figure 6: t-SNE visualization of PD vs Others.

making them easier to distinguish. In contrast, other cases show more scattered and heterogeneous feature distributions, which explains why some of them are more prone to being misclassified as PD.

## 5 CONCLUSION AND FUTURE WORK

**Conclusion.** This paper presents an end-to-end diagnostic framework for PD, termed PD-Diag-Net. To address the challenges of substantial inter-subject variability and heterogeneous data distributions in real clinical scenarios, the model first applies a unified pre-processing pipeline to all raw MRI data. It then incorporates two types of clinical priors (Relevance-Prior, Aging-Prior) and designs prior-guided feature aggregation and diagnosis modules, thereby significantly enhancing both robustness and interpretability. Extensive experimental results demonstrate that PD-Diag-Net consistently outperforms existing methods by a large margin on both normal and prodromal external test sets, underscoring its strong potential for clinical application.

**Future Work.** *On the technical side*, the heterogeneous feature distributions of "healthy" and "other neurological disorder" samples still lead to occasional misclassification as PD. Therefore, future work will prioritize expanding multi-hospital multimodal datasets (*e.g.*, MRI combined with clinical scales and behavioral signals) to reduce data imbalance problem. In parallel, optimizing the model architecture with advanced domain generalization and continual learning techniques to enhance robustness and generalization across diverse populations and imaging centers. *On the clinical side*, the current framework primarily addresses binary PD diagnosis, whereas actual clinical practice requires more fine-grained pathological reporting. Accordingly, future research will extend PD-Diag-Net to disease staging analysis, enabling not only the detection of PD but also the characterization of disease stage, the identification of pathological changes in specific brain regions, and the generation of personalized clinical recommendations. The ultimate goal is to develop a comprehensive intelligent diagnostic system that supports early screening, disease monitoring, and intervention planning, providing clinicians with more reliable, interpretable, and actionable assistance.

## 6 REPRODUCIBILITY STATEMENT

Our method is fully reproducible. The model architecture is illustrated in Fig. 3, and the hyper-parameters as well as training details (e.g., optimizer, learning rate, weight decay) are described in Sec. 4. Fig. 10 in the Appendix presents the complete loss curves during training, while Tab. 3 reports the training time across the three stages. In addition, both the source code and the curated dataset will be made publicly available in the future.

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

## A Appendix

## B The Use of Large Language Models (LLMs)

In this study, we leveraged the assistance of large language models (LLMs). Specifically, LLMs helped us polish and refine the language to better meet academic writing standards; during implementation, they assisted in checking and correcting code, thereby improving development efficiency and reliability; and in the model design stage, LLMs provided recommendations on backbone architectures, which facilitated a more efficient exploration of suitable network designs. These supports made our research process smoother and more effective.

## C Supplemented Related Work

While our main discussion has focused on AI-based approaches for PD analysis, it is also essential to contextualize PD within the broader neurological research landscape. Parkinson's disease (PD) is a progressive neurodegenerative disorder primarily driven by the degeneration of dopaminergic neurons in the substantia nigra, manifesting in core motor symptoms such as bradykinesia, rigidity, tremor, and postural instability (Bloem et al., 2021). In addition, PD encompasses a wide range of non-motor symptoms—including cognitive decline, mood disorders, autonomic dysfunction, and sleep disturbances—that often precede motor manifestations and profoundly affect patient quality of life (Chaudhuri et al., 2006). Neuropathological studies identify $\alpha$-synuclein aggregation and Lewy body formation as central hallmarks, placing PD within the broader class of synucleinopathies (Spillantini et al., 1997).

From the perspective of ongoing research, considerable effort has been devoted to identifying neuroimaging biomarkers and understanding spatiotemporal patterns of neurodegeneration. Structural MRI studies have highlighted cortical and subcortical atrophy patterns that differentiate PD from atypical parkinsonian syndromes (Uribe et al., 2018). Recent network-based approaches suggest that PD pathology may originate from epicenters within specific brain regions and then spread along large-scale brain networks, shaping heterogeneous clinical phenotypes (Zeighami et al., 2015). For example, connectome-based and longitudinal imaging studies demonstrate that distinct MRI epicenters can differentiate spatiotemporal trajectories of neurodegeneration in PD, thereby providing insight into disease heterogeneity and progression mechanisms (Duanmu et al., 2025).

Beyond structural imaging, multimodal studies integrate diffusion MRI, functional MRI, and molecular imaging to probe microstructural degeneration, network dysfunction, and dopaminergic deficits (Stoessl et al., 2011). These findings are increasingly leveraged to define prodromal PD, predict disease conversion, and improve differential diagnosis relative to other neurodegenerative disorders (Berg et al., 2015). Collectively, such neurological research provides the biological and clinical foundation upon which AI-based methods can build more robust, interpretable, and clinically useful models.

## D Full Diagnostic Flowchart of PD

The current diagnostic workflow for Parkinson's disease (PD) follows a stepwise process. First, neurologists conduct an initial clinical assessment based on the patient's medical history and hallmark motor symptoms, such as bradykinesia, resting tremor, and rigidity. For suspected cases, brain MRI is performed to exclude alternative structural or neurological conditions (e.g., stroke, tumors, hydrocephalus) that may mimic PD. Subsequently, the Movement Disorder Society (MDS) diagnostic criteria are applied. According to these criteria, the presence of any absolute exclusion factor rules out PD, whereas supportive features (e.g., a clear response to dopaminergic therapy, olfactory loss, characteristic imaging abnormalities) and red flags (e.g., rapid progression, early severe autonomic dysfunction, frequent falls, vertical gaze palsy) are weighed against each other. Patients are ultimately stratified into three categories: clinically established PD ($\geq 2$ supportive criteria, no red flags), clinically probable PD (red flags $\leq$ supportive criteria, no absolute exclusion), or not PD (if any absolute exclusion is present or red flags exceed supportive criteria). The flowchart is shown in Fig. 7.

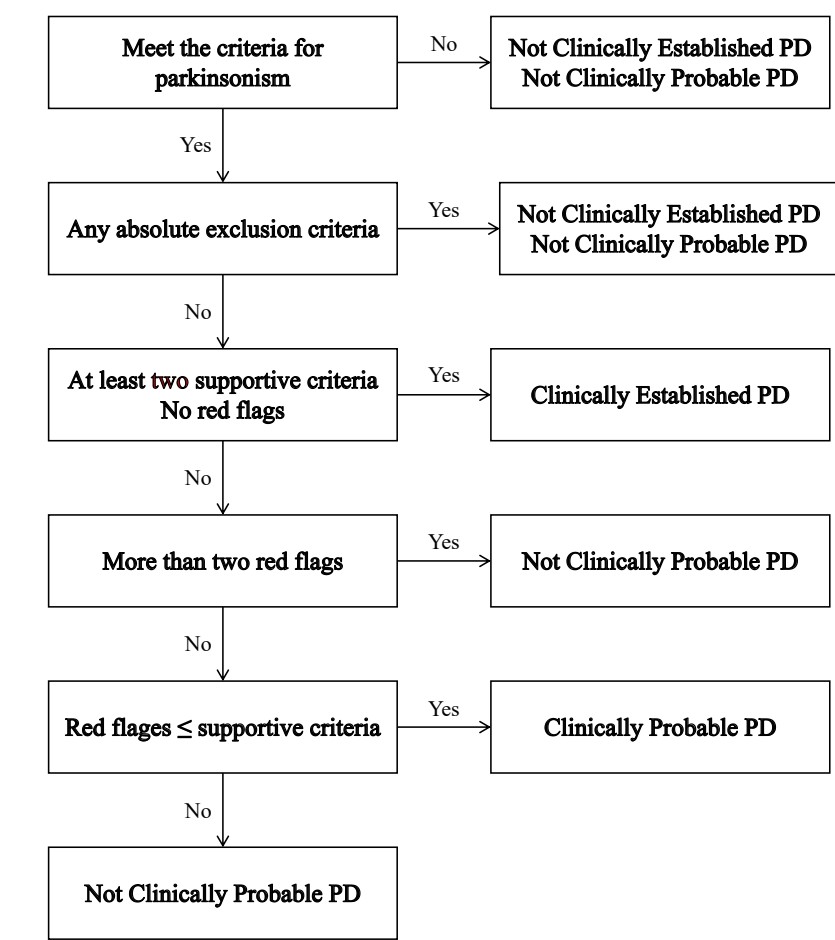

Figure 7: Full diagnostic flowchart of PD.

## E   BRAIN REGIONS AND THEIR RELEVANCE WITH PD

In the main text, we generated 48 brain regions based on the Harvard–Oxford atlas (Jenkinson et al., 2012) and classified their associations with PD according to the Relevance-Prior. A detailed description is provided in Tab. 5.

## F   DETAILS OF MRI PRE-PROCESSING MODULE

The MRI-Processor includes three steps: (1) Employ the HD-BET (Isensee et al., 2019) to automatically remove the skull and non-brain tissues, isolating the brain parenchyma and facilitating subsequent analysis. (2) Apply the Nonparametric Nonuniform intensity Normalization (N4) algorithm in ANTs (Avants et al., 2009) to correct bias fields, reducing intensity artifacts caused by magnetic field inhomogeneities during MRI acquisition and improving overall intensity uniformity. (3) Use the Symmetric Normalization (SyN) algorithm in ANTs to nonlinearly register individual MRI scans to the Montreal Neurological Institute (MNI) standard template, enabling consistent cross-subject comparison and group-level analysis. The flowchart is shown in Fig. 8, and the pseudo code is listed in Algorithm 1.

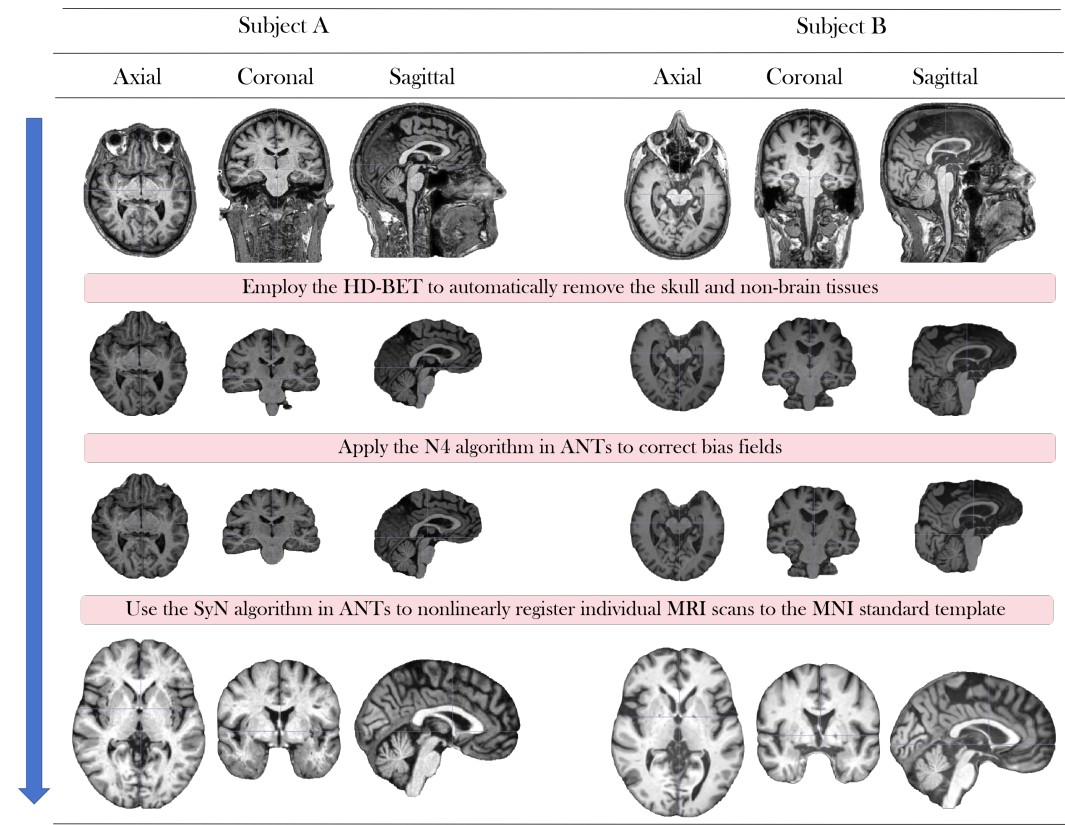

Figure 8: Flowchart of MRI-Processor.

---

**Algorithm 1** MRI-Processor Pipeline

---

**Require:** Raw MRI dataset, MNI template
**Ensure:** Preprocessed MRI aligned to MNI space
 1: **for** each subject in raw dataset **do**
 2:     **if** preprocessed result exists **then**
 3:         **continue**
 4:     **end if**
 5:     **Step 1: Skull stripping**
 6:     Run HD-BET on raw MRI → brain-extracted image
 7:     **Step 2: N4 bias correction**
 8:     Apply N4 correction → bias-corrected image
 9:     **Step 3: Registration**
10:     Register bias-corrected image to MNI template using ANTs (SyN)
11:     Save warped image to output directory
12:     Clean temporary files
13: **end for**

---

## G  SUPPLEMENTED EXPERIMENTS

### G.1  DETAILS OF EVALUATION METRIC

In this study, four primary evaluation metrics were employed to comprehensively assess the classification performance of the model. Before introducing these metrics, we first clarify the following basic concepts:

- **TP (True Positive)**: The number of positive samples (i.e., PD) correctly predicted as positive.
    $\Rightarrow$ Reflects the model's ability to correctly identify actual patients.
- **TN (True Negative)**: The number of negative samples (i.e., Others) correctly predicted as negative.
    $\Rightarrow$ Reflects the model's ability to correctly identify non-PD individuals.
- **FP (False Positive)**: The number of negative samples incorrectly predicted as positive.
    $\Rightarrow$ Represents non-PD individuals misdiagnosed as patients, potentially leading to overtreatment or unnecessary examinations.
- **FN (False Negative)**: The number of positive samples incorrectly predicted as negative.
    $\Rightarrow$ Represents true patients who were missed, which poses higher clinical risks.

Based on these definitions, we adopted the following evaluation metrics:

1. **Accuracy (ACC)**

$$\text{ACC} = \frac{TP + TN}{TP + TN + FP + FN} \tag{13}$$

Measures the overall proportion of correct predictions, reflecting the model's general performance.

2. **True Positive Rate (TPR, Sensitivity / Recall)**

$$\text{TPR} = \frac{TP}{TP + FN} \tag{14}$$

Indicates the proportion of actual patients correctly identified. A higher value means a lower miss rate.

3. **False Positive Rate (FPR)**

$$\text{FPR} = \frac{FP}{FP + TN} \tag{15}$$

Indicates the proportion of healthy individuals incorrectly classified as patients. A lower value reflects reduced misdiagnosis risk.

4. **Area Under the ROC Curve (AUC)** AUC refers to the area under the ROC curve, which plots TPR against FPR across different thresholds. Its value ranges from 0.5 to 1.0, with values closer to 1.0 indicating stronger ability to distinguish patients from healthy individuals.

### G.2 ROC CURVE

In the main text, we reported only ACC, TPR, and FPR in Tab. 1. Here, we additionally provide the ROC curve and the corresponding AUC value of our method, as shown in Fig. 9.

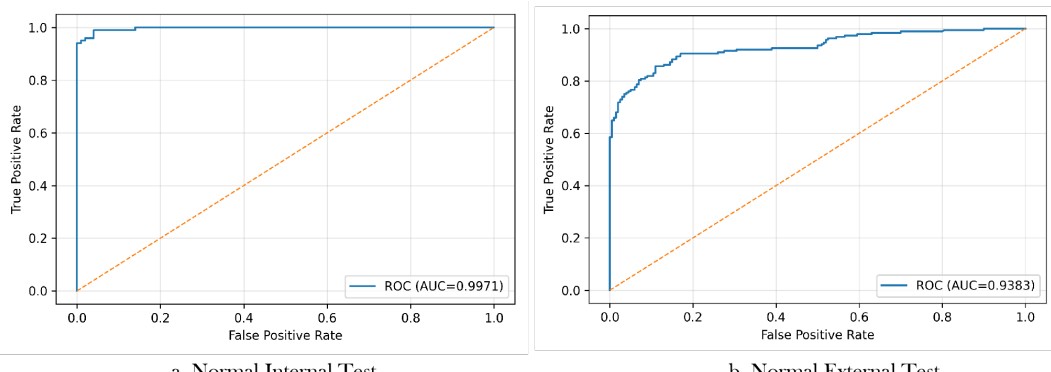

a. Normal Internal Test            b. Normal External Test

Figure 9: ROC Curve of normal internal and external tests.

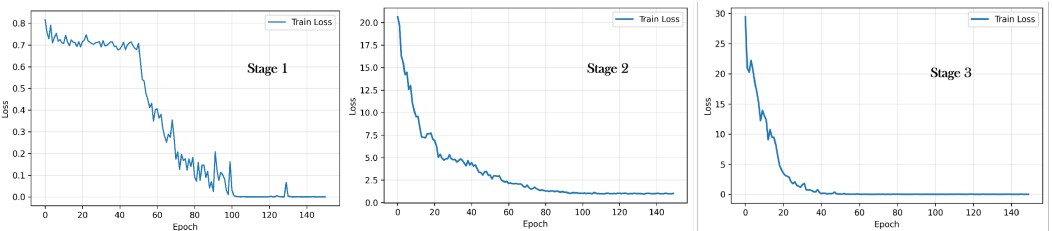

Figure 10: Training loss of three stages.

### G.3 TRAINING STRATEGY

Our framework consists of two branches and adopts a three-stage training strategy. Specifically, the feature extractor for dense representations is denoted as CNN1, the network in branch 1 as CNN2, and the network in branch 2 as CNN3. **Stage 1**: We jointly train CNN1 and CNN2 using data that includes both PD and Others categories. The networks are optimized with the classification loss in Eq. (7), and the learned parameters are saved. **Stage 2**: We fix CNN1 and train CNN3 using only healthy subjects from the Others group, under the assumption that their brain age in PD-associated regions equals their chronological age. CNN3 is optimized with a regression loss, and its parameters are saved. **Stage 3**: We load the parameters of CNN1, CNN2, and CNN3, and train on data including both PD and Others. The outputs of CNN3 are used to calibrate the classification logits, and the networks are fine-tuned with the final loss defined in Eq. (12). The loss trajectories across the three stages are illustrated in Fig. 10.

### G.4 TIME CONSUMING

All three stages were trained for 150 epochs on a single NVIDIA A100 GPU with a batch size of 4. The training and testing times are summarized in Tab. 3.

| Process | Time (minutes) |
|---|---|
| Training Stage 1 | 62 |
| Training Stage 2 | 65 |
| Training Stage 3 | 88 |
| Testing | 0.2 |

Table 3: Training and testing times on a single NVIDIA A100 GPU (batch size = 4).

### G.5 DIFFERENT BACKBONES.

In our study, feature extraction was performed using a 3D DenseNet. However, the choice of backbone is not particularly critical. We experimented with several advanced architectures and found that the results did not differ substantially. Here we replaced DenseNet with three other backbones, and the results on the normal external test set are summarized in Tab. 4.

| Backbone | ACC |
|---|---|
| S3D (Wald et al., 2025) | 86.3 |
| (Lee et al., 2022) | 86.5 |
| 3DMAE (Chen et al., 2023) | 85.7 |
| DenseNet (Ruiz et al., 2020) | 86.1 |

Table 4: Efficiency of different backbones on the normal external test set.

| Brain Region No. | Brain Region Name | Relevance with PD |
|---|---|---|
| 1 | Frontal Pole | None |
| 2 | Insular Cortex | Potentially |
| 3 | Superior Frontal Gyrus | Strong |
| 4 | Middle Frontal Gyrus | Strong |
| 5 | Inferior Frontal Gyrus, Triangular Part | Potentially |
| 6 | Inferior Frontal Gyrus, Opercular Part | Potentially |
| 7 | Precentral Gyrus | Strong |
| 8 | Temporal Pole | None |
| 9 | Superior Temporal Gyrus, Anterior Division | None |
| 10 | Superior Temporal Gyrus, Posterior Division | None |
| 11 | Middle Temporal Gyrus, Anterior Division | None |
| 12 | Middle Temporal Gyrus, Posterior Division | None |
| 13 | Temporooccipital Middle Temporal Gyrus | None |
| 14 | Inferior Temporal Gyrus, Anterior Division | None |
| 15 | Inferior Temporal Gyrus, Posterior Division | None |
| 16 | Temporooccipital Inferior Temporal Gyrus | None |
| 17 | Postcentral Gyrus | Potentially |
| 18 | Superior Parietal Lobule | Potentially |
| 19 | Supramarginal Gyrus, Anterior Division | None |
| 20 | Supramarginal Gyrus, Posterior Division | None |
| 21 | Angular Gyrus | Potentially |
| 22 | Lateral Occipital Cortex, Superior Division | None |
| 23 | Lateral Occipital Cortex, Inferior Division | None |
| 24 | Intracalcarine Cortex | None |
| 25 | Medial Frontal Cortex | Potentially |
| 26 | Juxtapositional Lobule Cortex (SMA) | Strong |
| 27 | Subcallosal Cortex | None |
| 28 | Paracingulate Gyrus | None |
| 29 | Anterior Cingulate Gyrus | None |
| 30 | Posterior Cingulate Gyrus | Potentially |
| 31 | Precuneous Cortex | Potentially |
| 32 | Cuneal Cortex | None |
| 33 | Orbitofrontal Cortex | None |
| 34 | Parahippocampal Gyrus, Anterior Division | None |
| 35 | Parahippocampal Gyrus, Posterior Division | None |
| 36 | Lingual Gyrus | None |
| 37 | Temporal Fusiform Cortex, Anterior Division | None |
| 38 | Temporal Fusiform Cortex, Posterior Division | None |
| 39 | Temporooccipital Fusiform Cortex | None |
| 40 | Occipital Fusiform Gyrus | None |
| 41 | Frontal Operculum Cortex | None |
| 42 | Central Opercular Cortex | None |
| 43 | Parietal Operculum Cortex | None |
| 44 | Planum Polare | None |
| 45 | Heschl's Gyrus | None |
| 46 | Planum Temporale | None |
| 47 | Supracalcarine Cortex | None |
| 48 | Occipital Pole | None |

Table 5: Brain regions and their relevance with PD.

