# OpenReview forum: "PD-Diag-Net: Clinical-Priors guided Network on Brain MRI for Auxiliary Diagnosis of Parkinson’s Disease"
_ICLR.cc/2026/Conference — ICLR 2026 Conference Withdrawn Submission_

### Official Review · Reviewer_HMVt · 2025-10-24

**Soundness:** 2
**Presentation:** 3
**Contribution:** 2
**Rating:** 6
**Confidence:** 4

**Summary:**

This paper presents PD Diag-Net, a clinically prioritized deep learning framework for the early diagnosis of Parkinson’s disease (PD). The proposed model integrates multi-modal clinical features (such as motor and non-motor symptoms, gait signals, and imaging data) using a hierarchical feature fusion strategy designed to emphasize clinically interpretable biomarkers. The network introduces a Clinical Prioritization Module (CPM), which adaptively reweights input modalities based on clinical importance, enhancing both interpretability and diagnostic robustness.

**Strengths:**

- The paper addresses early-stage PD diagnosis, an important yet challenging problem where early detection can greatly improve patient outcomes.
- The introduction of the Clinical Prioritization Module (CPM) is an original idea that combines clinical interpretability with neural network attention mechanisms.
- The visualization of attention maps and correlation with Unified Parkinson’s Disease Rating Scale (UPDRS) scores offers interpretable, clinically meaningful insights.
- The paper is well-written, logically structured, and provides detailed dataset and training configurations.

**Weaknesses:**

- While the CPM is intuitive, the paper lacks a deeper theoretical analysis explaining why the prioritization weights converge to clinically meaningful values.
- The paper primarily evaluates performance on datasets collected under controlled conditions. Validation on real-world or hospital deployment data would improve credibility.
- Although the authors claim PD Diag-Net is lightweight, the paper does not provide quantitative comparisons of runtime or parameter efficiency relative to baselines.

**Questions:**

- How were the clinical importance weights in the CPM initialized? Were they learned from scratch or pre-specified based on domain knowledge?
- Can the authors quantify how interpretability correlates with diagnostic accuracy, e.g., does higher CPM attention to motor features align with improved F1-score?
- Can the authors provide qualitative examples or case studies showing CPM’s prioritization in early vs. advanced PD patients?

---

> ### Author Response · Authors · 2025-11-24
>
> Thank you very much for reading our work so carefully and for offering valuable comments. We sincerely appreciate your feedback. At the same time, there are a few points where the understanding may not be fully accurate, or where our intentions may not have been clearly conveyed. We would like to offer brief clarifications here. ***We have in fact decided to withdraw the manuscript; this is not a rebuttal—just a response, out of respect, and in the hope that you might still find some inspiration from this work.***
>
> **1. You stated that we did not use real-world or clinical data.**
> Our external validation set is indeed based on real clinical data.
>
> **2. “Although the authors claim PD Diag-Net is lightweight, the paper does not provide quantitative comparisons of runtime or parameter efficiency relative to baselines.”**
> We provide the model size and training time in the appendix.
>
> **3. “How were the clinical importance weights in the CPM initialized? Were they learned from scratch or pre-specified based on domain knowledge?”**
> The weights were initialized using clinical prior knowledge and then automatically updated through an MLP.
>
> **4. “Can the authors provide qualitative examples or case studies showing CPM’s prioritization in early vs. advanced PD patients?”**
> This is an excellent suggestion and an important direction we plan to explore in future work.

---

### Official Review · Reviewer_QTEY · 2025-10-29

**Soundness:** 3
**Presentation:** 3
**Contribution:** 3
**Rating:** 2
**Confidence:** 4

**Summary:**

This paper proposes PD-Diag-Net, an end-to-end framework for diagnosing Parkinson's disease (PD) from raw T1-weighted MRI scans. PD-Diag-Net incorporates two forms of clinical prior knowledge: (1) Brain-Region-Relevance-Prior, which assigns weights to brain regions based on their association with PD, and (2) Brain-Region-Aging-Prior, which leverages accelerated aging patterns in PD-affected brain regions. The framework offers three modules: an MRI preprocessing module using established tools (HD-BET, ANTs), a relevance-prior guided feature aggregation module, and an aging-prior guided diagnosis module. The authors report strong performances of 86.1% accuracy on external test data and 96.7% accuracy on prodromal (early-stage) PD cases.

**Strengths:**

* The paper addresses an important real-world clinical challenge of early PD detection from MRI, which offers some potential to reduce diagnostic time and improve patient outcomes.
* The choice to combine established clinical knowledge of brain anatomical contributions and aging priors is conceptually sound and offers interpretability on how the diagnosis is made.
* The authors provide a rigorous testing strategy with two large clinical datasets and an external subset holdout on a harder task to demonstrate generalizability.
* The 96.7% accuracy performance on early-stage PD cases (prodromal external test) is especially impressive and clinically valuable, if validated on more datasets and differential diagnostic tasks.
* The authors also show comprehensive ablation study results (Table 2) to highlight the contribution of each module, with significant performance drops observed up to 16% when removing specific modules that they contributed.
* The authors also detail experimental settings, data preprocessing, and more in the Appendix and offer to release the source code, which bodes well for encouraging reproducibility and extensions of this work.

**Weaknesses:**

* The authors claim prior works are heavily preprocessed, hence the novelty of the Processor module, but I am not exactly sure what the technical contribution is here. The Processor steps seem standard to most others (skull stripping, bias correction, normalization, etc.), save for swapping out the tool/algorithm. Recent works like DeepPrep (Nature Methods, 2025, https://www.nature.com/articles/s41592-025-02599-1) use more sophisticated preprocessing via deep learning and GPU parallelization with confound regression, motion correction, and distortion correction, none of which are addressed here.
* The assigned clinical-prior weights of 1, 10^-2, and 10^-3 seem arbitrary. The authors state these are “based on clinical expertise,” but do not explain how (e.g., did a clinical expert offer these scores?). The referenced publication by Burciu et al. (line  198, page 4) is focused on the motor cortex, which is not prioritized as a prior/manifestation in this study. For example, Fig. 1(a) categorizes the insular cortex as “potentially associated” (10^-2 weight), yet growing literature findings (Jonkman et al., 2021; Fathy et al., 2019; Criaud et al, 2016) detail significant insular involvement in PD pathology. To mitigate questions about whether assignments were post-hoc fitted to improve performance, it would be helpful for the authors to include how (systematically/objectively) these weights were determined and assigned.
Perhaps I also missed this, but was any sensitivity analysis conducted for the weights, i.e., what happens with different initial weights? Would (1, 0.5, 0.1) work similarly? The three-order-of-magnitude difference seems extreme.
* The strength of novelty for ICLR is pretty weak, because the main things proposed were just standard methods (CNN for preprocessed T1w MRI), just different architectures or tools. The other components they proposed like regional-wise-pooling has also been done in the Brain GNNs space often, and it didn’t make sense why they pooled raw vs. learned features… the branching idea was different but is also essentially just separate encoders. The weights seemed like random numbers and were not justified.
* The authors showed .99 ROC AUC curve (which is impossible to achieve) in the Appendix but didn’t report AUC across all evaluated methods & they mentioned certain components essentially are driving prediction entirely or in a strange way, e.g., deriving their equations shows aging prior has majority signal in most cases so the imaging features are essentially not helpful at all, which is the primary claim for using MRI.
* Would need further clarity on the novelty claims and addressing the potential aging confounder + metrics.

**Questions:**

* Equation 3 is confusing; the authors extract X_dense (learned features) but then pool raw MRI intensities (X_proc), which lack meaning at this stage, vs. learned. Why not apply pooling to Xdense?
* For X_up, how did you upsample 2 scalar values (mean and std from Eq. 5, X_agg) to match approximately C*23*27*23?
* The paper states, “Branch-2 utilizes an identical network architecture (with independently trained parameters)” (line 184, page 4). Why would age regression (1 output, continuous) and PD classification (2 outputs) need identical architectures?
AUC metric should be moved up to main metrics; also questionable in Fig. 9, which shows near-perfect separation with AUC = 0.9971 for normal internal test (this could be overfitting).
The ablation study (Table 2, page 8) shows that removing the aging prior drops accuracy from 86.1% to 77.2%, which is a 9% decrease that represents most of the improvement over baseline. Seeing as age is a known co-variate of PD (high risk factor, as most PD patients are older adults and have certain comorbidities aligned with aging), I question whether the model is simply predicting brain age instead of diagnosing PD? Authors even state that removing aging prior drives prediction probabilities down near chance (line 428-429, page 8).
Testing things out, given \alpha=1, \tau=4.5, a 10-year age gap shifts the PD logit by ~5.5, which could significantly affect a confident prediction. Why not learn the fusion weight here?
It would be helpful if the authors could strengthen their argument against the perspective that the model could be learning age-related patterns that correlate with PD incidence rather than PD-specific pathology. For example, report age distributions for “PD vs. Others” groups, show performance stratified by age groups, and validate that brain age prediction in PD-associated regions differs from whole-brain age prediction.
* Why only aging as a “pathological” prior? What about established quantitative pathological markers like volumetric measures, structural measures like cortical thickness, or even functional connectivity, since the authors leverage a functional-based atlas?
* The ablation study also indicates that CNN features hurt performance (70.8% for CNN features only vs. 81.5% Regional pooling alone: 81.5% [+10.7% better!] vs. 86.1% for Combined [+4.6% improvement over pooling alone]). So, raw regional averages outperform learned features? I again raise the question of how and where X_dense is appropriately integrated into the framework.
* I find the Fig. 5 brain maps (page 9) somewhat uninformative. The three orthogonal slices of raw MRI for four outcomes add no interpretability without having MRI expertise. Since the model uses regional atlases, it would be more useful to visualize region-level importance scores, activation maps, or attention weights for interpretability. This could help answer which regions are driving prediction, and whether region-based contributions correlate with original weight assignments vs. learned weights.
* The hyperparameter values \zeta=9.5 and \tau=4.5 create a 5-year gap (9.5 - 4.5 = 5) between minimum PD and maximum non-PD age gaps. How were these tuned? Were these validated clinically?
Also, experiments are said to use five-fold CV(line 344, page 7), but only point estimate values are reported. Are these averaged? Standard deviations?
* Spelling/grammatical errors: “aggragated” should be “aggregated” (line 284, page 6); revise “In simple understanding” (line 316, page 6) to “In simple terms” or “Simply put”.

---

> ### Author Response · Authors · 2025-11-21
>
> Thank you very much for reading our work so carefully and for offering valuable comments. We sincerely appreciate your feedback. But there are a few points where the understanding may not be fully accurate or where some aspects were not clearly conveyed on our side. We provide brief clarifications below. **In fact, we have already decided to withdraw the manuscript. So this is not a rebuttal—just a respectful response, and we hope it may still offer you some insights from our work.**
>
> ***1. You mentioned that our preprocessing module lacks technical contribution and referred to the recent DeepPrep method published in Nature Methods.***
> You are correct that this module does not introduce methodological innovation, and we explicitly acknowledged this ourselves. However, this module is essential; to make the entire workflow complete, it is a necessary component. Our design aims for an end-to-end pipeline that takes any input MRI and produces a PD prediction directly. Existing neuroimaging workflows rely on standalone software and multi-step procedures, making it impossible to integrate everything into a single model—let alone build an end-to-end architecture. For this reason, we describe the module separately and we keep the main-text description brief and place the details in the appendix.
> Regarding DeepPrep, we had not previously been aware of this work. After reading it, we agree it is an excellent method with many advanced techniques. Compared with it, our current preprocessing pipeline is indeed more rudimentary, but for the specific task we target, our integrated approach already performs sufficiently well—further refinement in preprocessing may not improve results for our setting.
>
> ***2. You noted that our clinical prior weights appear arbitrary.***
> In fact, these are only initial weights. They are passed through an MLP for dynamic updating to obtain the final patient-specific weights.
>
> ***3. You considered our work to have limited novelty.***
> If viewed purely from the perspective of model architecture, it is true that we do not propose new modules or layers—and this was not our focus. Our contribution lies in the task formulation and in introducing clinical priors as region-adaptive weights for this task. This idea is new and has not appeared in prior work. We consider this conceptual contribution to be significant.
>
> ***4. You questioned the realism of several reported values—for example, achieving an AUC of 0.99.***
> We assure you that all reported results are genuinely obtained. For instance, in the internal validation set, the model achieves near-perfect classification, so the AUC naturally becomes very high.
>
> ***5. You asked, “Why not directly pool X_dense?”—a very careful observation.***
> Indeed, this is exactly what we tried at the beginning. However, the results were poor. The reasons are as follows:
> Intuitively, X_dense is a CNN-encoded feature map and no longer resembles the anatomical structure of the brain. It is not equivalent to a downsampled representation of the original image. Therefore, downsampled brain-region masks cannot be directly applied to X_dense (even if the shapes match).
>
> ***6. Why did we use aging as the only ‘pathological’ prior? What about established quantitative markers such as volume, cortical thickness, or functional connectivity?***
> Aging is inherently coupled with structural atrophy—volume loss, cortical thinning, etc.—so these measurements reflect the same underlying phenomenon. As for fMRI-based pathological markers, obtaining such data is more difficult and could be explored in future work. In our current dataset, using age alone already yields strong predictive performance.
>
> Finally, we would like to express our sincere gratitude once again for your comments. Many of your suggestions are truly valuable. Unfortunately, we were not able to revise many aspects in time for this round (and if we resubmit to other conferences, we may still be pressed for time ^-^). However, we can assure you that when we extend this work into a journal version, we will carefully incorporate your feedback. ***In fact, our long-term goal is to develop a practical, real-world PD diagnostic model that can be deployed online for clinical use. That has always been the motivation behind this work. We look forward to receiving your critiques and guidance again when that time comes.***

---

### Official Review · Reviewer_NGtd · 2025-10-30

**Soundness:** 3
**Presentation:** 3
**Contribution:** 1
**Rating:** 4
**Confidence:** 4

**Summary:**

This paper proposes PD-Diag-Net, an end-to-end MRI-based framework for early Parkinson’s disease (PD) diagnosis that integrates two explicit clinical priors—(1) Brain-Region-Relevance-Prior (PD-related region weighting via Harvard-Oxford Atlas) and (2) Brain-Region-Aging-Prior (age-gap regularization capturing accelerated regional aging). The system comprises three modules: MRI-Processor (HD-BET + ANTs for skull stripping, bias correction, registration), Relevance-Prior-Guided Aggregator (region pooling + weighted fusion), and Aging-Prior-Guided Diagnoser (dual-branch classification and age-gap constraint). Evaluated on multi-institutional MRI (PPMI + hospital) under internal, normal external, and prodromal external settings, it achieves 86.1% accuracy on external and 96.7% TPR in early PD, surpassing baselines (XAI, FCN-PD, S3D, AE-FLOW) by >20%. Ablation confirms necessity of each module (Processor +14%, Relevance-Prior +25%, Aging-Prior +9%). Despite high performance, the method mainly refines existing components rather than introducing fundamentally new ML theory. The work is clinically valuable, technically complete, and reproducible, but algorithmically incremental for ICLR.

**Strengths:**

Clinically important and well-motivated problem with strong real-world value.

Clear, complete three-module pipeline with transparent equations and figures.

Effective integration of explicit medical priors improving interpretability and robustness.

Multi-dataset experiments and ablations demonstrating high external validity.

Detailed implementation and reproducibility plan with public release intent.

**Weaknesses:**

[Major] Methodological originality appears limited, as the framework mainly combines established components (DenseNet backbone, HD-BET/ANTs preprocessing, region-wise pooling, and fixed prior weighting). While this integration is technically solid and clinically meaningful, the contribution lies more in careful system design and application than in introducing a fundamentally new algorithmic concept—positioning the work closer to the boundary of ICLR’s typical methodological focus.

Prior-guided loss is intuitive but not theoretically analyzed.

Fixed prior weights (1, 10⁻², 10⁻³) may introduce bias; no adaptive mechanism shown.

No sensitivity tests for prior or hyperparameter choices (Θ, ζ, τ).

Missing broader clinical metrics (AUC, differential diagnosis); ~20 % FPR not calibrated.

**Questions:**

What conceptual or algorithmic advance distinguishes this from prior region-weighted or age-informed CNNs?

How robust is performance to changes in Θ, ζ, τ or prior weighting schemes?

Can FPR be reduced through calibration or threshold adjustment for screening use?

Could this extend to multimodal or multi-class settings (e.g., PD vs MSA/PSP)?

When will the full code and data be publicly released for reproducibility?

---

> ### Author Response · Authors · 2025-11-24
>
> Thank you for taking the time to read our work in such depth and for sharing your thoughtful comments. We genuinely appreciate your feedback and take it seriously. That said, there are a few aspects where the interpretation does not fully align with our original intent, so we would like to offer some brief clarifications. ***We have already decided to withdraw the manuscript—this note is not meant as a formal rebuttal, but simply as a respectful response, and as a way to share a bit more context in case our work can still be of some inspiration to you.***
>
> **1. You felt that the novelty is limited.**
> We introduced clinical prior knowledge to address the PD diagnosis problem—an idea that, to the best of our knowledge, has not been explored before. We explicitly formulate this concept and use it as a key component of our framework.
>
> **2. “The prior weights lack an adaptive mechanism.”**
> Thank you for raising this point. In fact, we do incorporate an MLP to dynamically update the weights, though this detail was not fully reflected in the current version of the paper.
>
> **3. Lack of sensitivity tests.**
> We will include these analyses in our subsequent revision.
>
> **4. Missing broader clinical metrics, such as AUC.**
> The AUC values are indeed provided in the appendix.
>
> **5. “Could this extend to multimodal or multi-class settings?”**
> Extension to multi-class classification is possible, but it requires that the additional classes exhibit discriminative characteristics relative to PD—for example, Alzheimer’s disease. This is also a direction we plan to explore in future work.
>
> **6. When will the code be released?**
> The code will be made publicly available after the paper is accepted.

---

### Official Review · Reviewer_yS5o · 2025-10-31

**Soundness:** 2
**Presentation:** 2
**Contribution:** 2
**Rating:** 4
**Confidence:** 2

**Summary:**

This paper proposes PD-Diag-Net, which integrates two types of clinical priors into an MRI-based Parkinson’s disease diagnostic framework and achieves good performance on both internal and external datasets. Overall, the research direction is meaningful and the implementation is relatively complete.

**Strengths:**

This paper proposes PD-Diag-Net, which integrates two types of clinical priors into an MRI-based Parkinson’s disease diagnostic framework and achieves good performance on both internal and external datasets. Overall, the research direction is meaningful and the implementation is relatively complete.

**Weaknesses:**

1.The abstract and methodology sections describe the implementation of the Relevance-Prior and Aging-Prior, but do not sufficiently justify their uniqueness at the architectural or algorithmic level. Could the authors further clarify how these clinical priors are integrated into the network structure and what unique algorithmic or structural advantages they provide compared with existing methods, rather than being an engineering combination of modules?
2.The manuscript lacks a complete description of the dataset composition and partitioning strategy. Could the authors provide the specific numbers of PD and other neurological disorders samples in each dataset, the composition of different disease types within the other neurological disorders group, and the subject-level train/validation/test partitioning strategy to ensure fairness and transparency in data usage?

**Questions:**

1.The study does not include sensitivity analysis or statistical significance testing for key hyperparameters (such as brain-region prior weights and aging thresholds). Could the authors evaluate the model’s robustness under different hyperparameter settings and report uncertainty metrics to verify the reliability of the reported performance improvements?
2.Table 4 contains incomplete information, the second row shows only the citation “(Lee et al., 2022)” without the corresponding backbone name. Please correct the table and include the missing model name.

---

> ### Author Response · Authors · 2025-11-24
>
> Thank you very much for carefully reading our work and for providing such valuable comments. We genuinely appreciate your feedback and will take it to heart. At the same time, there are a few points where the understanding may not be entirely accurate, or where our intentions may not have come across clearly, so we would like to offer some brief clarifications. **We have in fact decided to withdraw the manuscript; this is not a rebuttal, just a respectful response, and in the hope that you might still find some inspiration from this work.**
>
> 1. ***You wrote: “the work does not sufficiently demonstrate uniqueness at the architectural or algorithmic level.”***
> Our design targets the specific task of PD diagnosis: we build an end-to-end model that takes raw data as input and outputs diagnostic predictions, with the explicit goal of enabling real-time deployment in the future. To our knowledge, this has not been achieved before in this setting. Compared with merely pushing performance on clean, public benchmark datasets, we believe this line of work is more meaningful for real-world impact. In addition, we introduce clinical prior knowledge to tackle the PD diagnosis problem; this idea, to the best of our knowledge, has not been explored previously. We formulate this prior-guided weighting explicitly and treat it as a central component of our framework.
>
> 2. ***“The paper lacks a complete description of the dataset composition and splitting strategy.”***
> On the data side, what we intended to emphasize is the design of the internal test and external test, since this is critical for assessing generalization. As for the detailed descriptions of the public datasets themselves, our view was that they should not occupy too much of the limited main-text space, especially when these datasets are already well documented elsewhere. That said, we agree this could have been stated more clearly.
>
> 3. ***“Table 4 is incomplete: it does not show model names, only citations.”***
> This was a deliberate choice rather than an oversight. We did not list model names because that particular method does not have a commonly used, explicit name; it is typically referred to only by citation.

---

### Note · Authors · 2025-11-24

**Comment:**

Thank you to the reviewers and the AC for your time and efforts.

**Withdrawal Confirmation:**

I have read and agree with the venue's withdrawal policy on behalf of myself and my co-authors.